# SYK-623, a δ Opioid Receptor Inverse Agonist, Mitigates Chronic Stress-Induced Behavioral Abnormalities and Disrupted Neurogenesis

**DOI:** 10.3390/jcm13020608

**Published:** 2024-01-21

**Authors:** Takashi Iwai, Rei Mishima, Shigeto Hirayama, Honoka Nakajima, Misa Oyama, Shun Watanabe, Hideaki Fujii, Mitsuo Tanabe

**Affiliations:** 1Laboratory of Pharmacology, School of Pharmacy, Kitasato University, 5-9-1 Shirokane, Minato-ku, Tokyo 108-8641, Japan; iwait@pharm.kitasato-u.ac.jp (T.I.); r.mishima@ono-pharma.com (R.M.); zatta.midios01@gmail.com (H.N.); oyamam@pharm.kitasato-u.ac.jp (M.O.); watanabes@pharm.kitasato-u.ac.jp (S.W.); 2Medicinal Research Laboratories, School of Pharmacy, Kitasato University, 5-9-1 Shirokane, Minato-ku, Tokyo 108-8641, Japan; hirayamas@pharm.kitasato-u.ac.jp (S.H.); fujiih@pharm.kitasato-u.ac.jp (H.F.); 3Laboratory of Medicinal Chemistry, School of Pharmacy, Kitasato University, 5-9-1 Shirokane, Minato-ku, Tokyo 108-8641, Japan

**Keywords:** delta opioid, stress, learning and memory, depression, neurogenesis, GABA

## Abstract

The δ opioid receptor (DOR) inverse agonist has been demonstrated to improve learning and memory impairment in mice subjected to restraint stress. Here, we investigated the effects of SYK-623, a new DOR inverse agonist, on behavioral, immunohistochemical, and biochemical abnormalities in a mouse model of imipramine treatment-resistant depression. Male ddY mice received daily treatment of adrenocorticotropic hormone (ACTH) combined with chronic mild stress exposure (ACMS). SYK-623, imipramine, or the vehicle was administered once daily before ACMS. After three weeks, ACMS mice showed impaired learning and memory in the Y-maze test and increased immobility time in the forced swim test. SYK-623, but not imipramine, significantly suppressed behavioral abnormalities caused by ACMS. Based on the fluorescent immunohistochemical analysis of the hippocampus, ACMS induced a reduction in astrocytes and newborn neurons, similar to the reported findings observed in the postmortem brains of depressed patients. In addition, the number of parvalbumin-positive GABA neurons, which play a crucial role in neurogenesis, was reduced in the hippocampus, and western blot analysis showed decreased glutamic acid decarboxylase protein levels. These changes, except for the decrease in astrocytes, were suppressed by SYK-623. Thus, SYK-623 mitigates behavioral abnormalities and disturbed neurogenesis caused by chronic stress.

## 1. Introduction

Depression is a prevalent disorder often precipitated by chronic stress, which not only causes mood disturbances but also impairs cognitive function [1]. Accumulating evidence indicates that experiencing stressful events affects cognitive function [2,3] and that perceived stress levels are associated with cognitive impairment in humans [4]. In a preclinical study, chronic stress exposure in rodents resulted in depression-like behavior and a decline in cognitive function [5]. Chronic stress paradigms, exemplified by chronic mild stress (CMS), are commonly used to establish depression model mice [6]. Cognitive impairment in depression is difficult to treat with most antidepressants and may even be exacerbated by tricyclic antidepressants, perhaps because of their anticholinergic effects [7]. However, cognitive dysfunction in conventionally used chronic stress models is improved by imipramine [8,9,10,11]. This suggests that chronic stress models produced by conventional protocols may not be sufficiently accurate to predict the effects of drugs on cognitive dysfunction in depression.

The δ opioid receptor (DOR) is a subtype of the opioid receptor. In the brain, DORs are abundantly distributed in areas involved in emotions, such as the cerebral cortex, striatum, lateral nucleus, hippocampus, and amygdala [12]. As indicated by the distribution, DOR agonists are known to have anti-anxiety and anti-depressant effects [13]. Conversely, DOR antagonists not only exhibit harmful effects, such as anxiety induction, but they also show beneficial effects, such as antitussive actions [14,15]. Despite their attractive pharmacological profiles, DOR ligands have not been developed for practical use. Several DOR agonists were evaluated in clinical trials for postherpetic neuralgia (NCT01058642), post-molar extraction pain (NCT00993863), osteoarthritis of the knee (NCT00979953), and anxious depression (NCT00759395), but they did not demonstrate sufficient efficacy. 

Recently, we synthesized SYK-657, a potent small-molecule DOR inverse agonist, and SYK-623, which has an increased affinity for DORs compared to the former [16,17]. Receptors may have intrinsic activity even in the absence of agonist binding. Agonists bind to receptors and increase their activity, whereas inverse agonists decrease the intrinsic activity of receptors (negative intrinsic activity) [18]. SYK-623 and SYK-657 ameliorated spatial working memory deficits induced by repetitive restraint stress in mice, which were inhibited by the DOR antagonist naltrindole (NTI) [16,19]. These results provide hope for the efficacy of DOR inverse agonists in treating cognitive decline in chronic stress-induced psychiatric disorders but should also be verified in different chronic stress models. 

In this study, we employed an imipramine-resistant stress mouse model, which was treated with repeated administration of adrenocorticotropic hormone (ACTH) and CMS, to investigate the effects of SYK-623 on depression-like behavior and cognitive decline induced by chronic stress. Previously, it was reported that rats and mice that were chronically administered ACTH were insensitive to tricyclic antidepressants in a forced swimming test [20,21]. However, ACTH did not induce depression-like behavioral changes. Therefore, we demonstrated that the combination of repeated ACTH administration with CMS to induce depression-like behavior and cognitive decline resulted in symptoms of imipramine resistance, as is discussed later in the manuscript: We abbreviated the combined treatment of ACTH and CMS as ACMS to distinguish it from the conventional CMS. Chronic stress reduces astrocytes, neurogenesis, and parvalbumin (PV)-positive GABA neurons, which are crucial for the expression of learning, memory, and antidepressant effects [22,23,24]. Thus, to investigate the mechanism of the effects of SYK-623, we examined whether SYK-623 protects these cells against ACMS in the hippocampus.

## 2. Materials and Methods

### 2.1. Subjects

All experiments were approved by the Institutional Animal Care and Use Committee of Kitasato University (ethical approval code: 17-17, 21-6) and were performed in accordance with the guidelines of the National Institutes of Health and the Japanese Pharmacological Society. Six-week-old male ddY mice (Japan SLC, Shizuoka, Japan) were used for all the experiments, since we previously reported that repeated administration of ACTH causes imipramine resistance in ddY mice [21]. Treatment groupings are shown in Appendix A. The ddY strain is an outbred one, and it has been maintained as a closed colony. This strain, along with the ICR strain, has been widely used in drug efficacy trials and various fields of research in Japan. Mice were kept in a controlled environment, with controlled lighting (12 h light/dark cycle, lights on from 08:00 to 20:00) and temperature (23 ± 1 °C), and they were given free access to food and water. Mice were housed at 4–5 mice per cage (30 × 20 × 13 cm). Bedding material was made of paper (paperclean^®^, Japan SLC) and was replaced after the wet bedding stress of ACMS (Table 1). To avoid the impact of various environmental changes on the mice, experiments were conducted in a dedicated soundproof room until the completion of the behavioral experiments.

### 2.2. ACMS Model Mice

The experimental schedule is presented in Table 1. ACMS was performed with a daily combination treatment of ACTH and CMS, as described in the Introduction. Individual stressors in CMS are exposed to animals at various timings and methods, depending on the laboratory [25]. To avoid excessive impairments induced by the combination of ACTH, we used a CMS protocol, which did cause moderate weight loss in the preliminary experiments. To elevate low corticosterone levels in the light phase, ACTH (Cortrosyn, 0.45 mg/kg, Daiichi Sankyo, Tokyo, Japan) was subcutaneously administered once daily at 9:30–11:30 based on previous reports [21]. The CMS protocol consisted of one or two daily exposures to random stressors between 9:30–14:00 (short stressor) and overnight (overnight stressor), including the following: (1) 2 h of restraint in a 50 mL plastic tube (Falcon) with openings for ventilation on both sides, (2) 15 min of forced swimming (28 °C), (3) wet bedding, (4) night lighting, (5) cage tilting from horizontal to 45°, (6) water deprivation, and (7) food deprivation. All stressors were repeated throughout the 31-day experiment.

### 2.3. Drugs

SYK-623 and naltrindole (NTI) were synthesized by Fujii et al. (Kitasato University, Tokyo, Japan). Imipramine was purchased from Sigma-Aldrich (St. Louis, MO, USA). SYK-623 was dissolved in dimethylsulfoxide at 2 mg/mL and stored at −30 °C. At the time of use, it was diluted 10-fold by saline (Otsuka Pharmaceutical Co. Ltd., Tokyo, Japan) and administered intraperitoneally (i.p.) to mice at 2 mg/kg. Imipramine (30 mg/kg, i.p.) and NTI (2 mg/kg, i.p.) were dissolved in saline (Otsuka Pharmaceutical Co. Ltd.) at the time of use and administered to mice. These drugs were administered daily immediately prior to ACTH treatment (Table 1). DMSO-containing (10%) saline and normal saline were administered as vehicle controls for SYK-623 and other drugs (imipramine and NTI), respectively.

### 2.4. Tail Suspension Test

The tail suspension test was performed as previously described [26]. Mice were individually suspended from the ceiling of an open front box (distance from floor, 35 cm) using adhesive tape affixed 2 cm from the tip of the tail. The mice demonstrated several escape-oriented behaviors interspersed with bouts of immobility as the sessions progressed. A 5 min test session was videotaped, and the immobility time was measured as the time at which the mice were judged to cease escape-motivated behaviors.

### 2.5. Spontaneous Alternation Test (Y-Maze)

The spontaneous alternation test was performed using Y-maze as previously reported [27]. Each mouse was placed at the end of one arm and allowed to move freely through the maze during a 5 min session. The arm entries were recorded visually by a blinded experimenter. Spontaneous alternation behavior was defined as successive entries into the three arms of overlapping triplet sets. The effect was calculated as the percentage of alternation according to the following formula: % alternation = {(number of alternations)/(total number of arm entries − 2)} × 100 (%). For example, if the sequence of arm entry in a single 5 min session was ABCBABCACBAC, then the experimenter would score seven spontaneous alternations (in order: ABC, CBA, ABC, BCA, ACB, CBA, BAC). With a total of 12 arm entries, the spontaneous % alternation would be 70%.

### 2.6. Novel Arm Recognition Test (Modified Y-Maze)

A modified version of the Y-maze was used for novelty arm recognition, as previously reported [28]. The test consisted of two parts: a training session and a test session. During the training sessions, the entrance to one of the three arms was closed with a wall. The mice were allowed to explore the other two arms freely for 10 min. The test session was conducted two hours after the training session. The closed arm was opened and defined as the novel arm. The mice were allowed to explore the apparatus for 5 min. The percentage of entries in the novel arm was calculated based on the total number of entries.

### 2.7. Novel Object Recognition Tests

The novel object recognition test was performed in an open field, with each side measuring 30 cm as described previously [29]. The test consisted of two parts: a training session and a test session. In the training session, mice were allowed to explore a field in which two identical objects were placed for 10 min. The test session was performed 24 h after the training session, in which mice were allowed to explore an open field for 5 min, but with one of the original objects replaced by another (novel object). The duration for which each mouse oriented its nose towards each object was recorded. The two objects are plastic toys in which the mice showed equal interest in the preliminary experiments. The time spent exploring the novel object was calculated as a percentage of total object exploration time.

### 2.8. Open Field Test

The open-field test was performed using a modified version of the procedure described in our previous report [27]. The open-field apparatus consisted of a square area (50 × 50 cm) with opaque walls that were 50 cm in height. Mice were placed in a corner of the open field facing the opaque walls and were allowed to explore freely for 5 min. The traveling distance was measured by using a SMART video tracking system (Panlab, Harvard Apparatus, Holliston, MA, USA).

### 2.9. Immunohistochemistry

After the ACMS exposure for 31 days, mice were perfused with 4% paraformaldehyde (PFA). The isolated brains were post-fixed in PFA overnight and then sequentially incubated in 15% and 30% sucrose in PBS. The brains were sliced into 30 μm thick horizontal sections. The sections were heated in a citrate buffer (pH 6) at 80 °C for 20 min and then immersed in PBS with 0.1% Triton X-100 for 30 min. Blocking was performed in 5% bovine serum albumin-containing PBS with Tween 20 (PBST) for 30 min. The sections were immersed in the primary antibody solution [goat anti-doublecortin (DCX) antibody (1:50, Santa Cruz Biotechnology, Dallas, TX, USA), mouse anti-GFAP antibody (1:500, Sigma); rabbit anti-PV antibody (1:1000, Abcam, Cambridge, UK)] overnight at room temperature (approximately 25 °C). After three washes with PBST, the sections were incubated for 1 h in a mixture of diluted secondary antibodies [CF568-conjugated donkey anti-mouse IgG and CF633-conjugated donkey anti-goat IgG (1:1000, Biotium, Heyward, CA, USA), Alexa Fluor 488-conjugated goat anti-rabbit IgG (1:750, Jackson ImmunoResearch Laboratories, Inc., West Grove, PA, USA)] and DAPI (1 μg/mL, Dojindo, Osaka, Japan). After three washes with PBST, the sections were mounted using mounting medium. Immunofluorescent images were obtained using an LSM 710 confocal laser scanning microscope (Zeiss, Göttingen, Germany). The GFAP- or DCX-positive areas and the number of PV-positive cells were measured using Fiji/ImageJ software (v1.53) [30].

### 2.10. SDS-PAGE and Western Blotting

Hippocampal tissue samples were homogenized in lysis buffer [50 mM Tris (pH 7.5), 150 mM NaCl, 2% Triton X-100, 0.25% *v*/*v*, and protease inhibitor cocktail (Nacalai Tesque, Kyoto, Japan)]. The protein content was quantified using the bicinchoninic acid protein assay method (Nacalai Tesque). The tissue samples (30 μg/lane) were separated on a 10% SDS-polyacrylamide gel. Following transfer to polyvinylidene difluoride (PVDF) membranes using the wet method, the blocking step was performed for 60 min with Tris-buffered saline (TBS) containing 5% nonfat dried milk and 0.1% Tween 20 at room temperature and then probed with anti-GAD67 antibody (1:10,000, Abcam) or anti-β-actin antibody (1:20,000, Genetex, Irvine, CA, USA) at room temperature for 1 h. After three washes with TBS containing 0.05% Tween 20, the membranes were incubated with horseradish peroxidase-conjugated anti-mouse IgG (Jackson ImmunoResearch Laboratories) diluted 1:10,000 in TBS containing 0.05% Tween 20 at room temperature for 1 h, followed by detection using enhanced chemiluminescence (Chemi-Lumi One, Nacalai Tesque). Band intensities were semi-quantified using a CS Analyzer 3.0 (ATTO Corporation, Tokyo, Japan).

### 2.11. Statistics

Data are expressed as mean ± SEM. All statistical analyses were performed using GraphPad Prism 10 (GraphPad Software, San Diego, CA, USA), and the results are indicated in the figure legends. The criterion for significance was set at *p* < 0.05. Two-way analysis of variance was performed for the time course of body weight changes, and one-way analysis of variance was performed for behavioral experiments, immunohistochemistry, and Western blotting. Holm-Sidak’s method, which exhibits adequate type I error rate control and high power, was used for all multiple comparisons [31].

## 3. Results

### 3.1. ACMS-Induced Imipramine-Resistant Physical and Behavioral Impairments

Chronic stress exposure to rodents is generally accompanied by weight loss, adrenal enlargement, and thymus atrophy [32]. We observed that ACMS mice had significantly reduced body weight gain compared to control mice that received only vehicle administration (Figure 1A) and that adrenal hypertrophy (Figure 1B) and thymic atrophy (Figure 1C) occurred after 4 weeks, confirming that ACMS induced physical symptoms characteristic of chronic stress in mice. Imipramine had no significant effect on ACMS-induced physical changes (Figure 1A–C).

To examine the effects of imipramine on depression-like behavior in ACMS mice, a tail suspension test was performed. ACMS mice showed prolonged immobility time compared to control mice, which was not prevented by imipramine treatment (Figure 1D). To examine the effects of imipramine on the impairment of working memory and short-term memory in ACMS mice, a spontaneous alternation behavior test and novel arm recognition test were performed using a Y-maze apparatus. In the spontaneous alternation behavior test, ACMS mice showed a significantly lower percentage of alternation than control mice, which was not prevented by imipramine treatment (Figure 1E). In the novel arm recognition test, ACMS mice showed a lower percentage of entries into the novel arm compared to control mice, which was not prevented by imipramine treatment (Figure 1F).

To examine the effects of imipramine on long-term memory impairment, we performed the novel object recognition test. ACMS mice showed a significantly lower % discrimination ratio for the novel object than control mice, which was not prevented by imipramine treatment (Figure 1G). Spontaneous locomotor activity in the open field test was not significantly different between the groups (Figure 1H). These results indicate that the ACMS induces imipramine-resistant depressive behavior and cognitive dysfunction.

### 3.2. DOR Inverse Agonist SYK-623 Prevented Induction of Behavioral Impairment in ACMS Mice

To examine whether SYK-623 prevented the induction of impairment caused by ACMS, SYK-623 was administered concurrently with ACMS. SYK-623 had no significant effect on ACMS-induced loss of body weight (Figure 2A), adrenal hypertrophy (Figure 2B), or thymic atrophy (Figure 2C). In the tail suspension test, SYK-623 significantly attenuated the ACMS-induced prolongation of immobility time (Figure 2D), indicating that SYK-623 has antidepressant-like effects. In addition, SYK-623 significantly attenuated the ACMS-induced decline in % alternation (Figure 2E) and % entries into the novel arm (Figure 2F), indicating that SYK-623 protected working memory and short-term memory functions from chronic stress. In contrast, SYK-623 did not attenuate the ACMS-induced decline in the % discrimination ratio (Figure 2G), indicating that SYK-623 does not protect long-term memory function from chronic stress. Spontaneous locomotor activity was not significantly different between the groups (Figure 2H). These results suggest that SYK-623 protects short-term cognitive function, including working memory.

### 3.3. The DOR Neutral Antagonist NTI Did Not Suppress ACMS-Induced Behavioral Impairment

To examine the possibility that the effect of SYK-623 on ACMS may be due to the antagonism of the endogenous ligand, we examined the effect of chronic treatment with NTI, a neutral antagonist that was administered to mice in parallel with ACMS. NTI did not significantly affect ACMS-induced weight loss (Figure 3A), adrenal hypertrophy (Figure 3B), or thymic atrophy (Figure 3C). NTI did not suppress ACMS-induced prolongation of immobility time but rather tended to exacerbate it (Figure 3D, *p* = 0.055, vs. ASMC + saline), indicating that chronic treatment with a DOR neutral antagonist did not show antidepressant-like effects. NTI did not attenuate the ACMS-induced decline in % alternation (Figure 3E) or % entries into the novel arm (Figure 3F), indicating that NTI did not protect working memory and short-term memory functions from chronic stress. Similarly, NTI did not attenuate the ACMS-induced decline in the percentage discrimination ratio in the novel object recognition test (Figure 3G), indicating that NTI does not protect long-term memory function from chronic stress. Spontaneous locomotor activity was not significantly different between the groups (Figure 3H). These results suggest that neutral antagonists do not exhibit anti-stress effects, unlike SYK-623.

### 3.4. SYK-623 Does Not Suppress ACMS-Induced Imipramine-Resistant Astrocyte Loss in the Hippocampus

Astrocyte loss has been reported in the brains of patients with depression and chronically stressed mice and is reversed by antidepressants, including imipramine, in CMS models [22,33]. To determine whether ACMS induces astrocytic loss and imipramine resistance, fluorescent immunohistochemistry for GFAP, a marker of astrocytes, was performed using frozen sections of the hippocampus. ACMS mice showed decreased GFAP-positive areas compared to control mice, which were not affected by imipramine treatment in CA1 (Figure 4A, 1st row and Figure 4B, upper left) and the dentate gyrus (Figure 4A, 2nd row and Figure 4B, upper right) of the hippocampus. Next, we determined whether SYK-623 suppressed ACMS-induced imipramine-resistant astrocyte loss. The ACMS-induced decrease in GFAP-positive areas was not affected by SYK-623 treatment in CA1 (Figure 4A, 3rd row and Figure 4B, lower left) and the dentate gyrus (Figure 4A, 4th row and Figure 4B, lower right) of the hippocampus. 

### 3.5. SYK-623 Suppressed ACMS-Induced Imipramine-Resistant Decrease in Neurogenesis

Decreased hippocampal neurogenesis is considered a major cause of hippocampal dysfunction due to chronic stress [5,23,34]. The effects of chronic stress and drugs on neurogenesis can be detected by a decrease or increase in DCX, a marker of immature neurons [35,36,37]. To determine whether ACMS reduced neurogenesis, fluorescent immunohistochemistry for DCX was performed in the hippocampal dentate gyrus. ACMS mice had significantly reduced DCX-positive areas in the dentate gyrus of the hippocampus compared with control mice, which were not significantly suppressed by imipramine treatment (Figure 5A, upper panel; Figure 5B, left panel). Next, we determined whether SYK-623 suppressed the ACMS-induced imipramine-resistant decrease in neurogenesis. ACMS did not decrease the number of DCX-positive areas in the SYK-623-treated mice (Figure 5A, lower panel and Figure 5B, right panel). These results indicate that SYK-623 protects neurogenesis from chronic stress.

### 3.6. SYK-623 Suppresses ACMS-Induced Impairment of GABA Neurons

GABA neurons in the hippocampal dentate gyrus play an important role in neurogenesis and have been reported to be impaired by chronic stress [24]. To determine whether SYK-623 protects GABA neurons from ACMS, we counted PV-positive GABA (PV-GABA) neurons, a type of GABA neuron. ACMS mice showed a significantly reduced density of PV-positive cells compared to control mice, which was significantly suppressed by SYK-623 treatment (Figure 6A,C). Next, the protein levels of the GABA neuron marker GAD67 were measured in the hippocampal tissue by western blotting. ACMS mice showed significantly reduced hippocampal GAD67 levels, which were significantly suppressed by SYK-623 treatment (Figure 6B,D). These results indicate that SYK-623 protects PV-GABA neurons from chronic stress. 

## 4. Discussion

Previously, we showed that the acute administration of DOR inverse agonists ameliorated impairments in spatial working memory due to chronic restraint stress, and we predicted that DOR inverse agonists would have a beneficial effect on cognitive impairment in chronic stress disorders, such as depression [16]. However, in this mouse model, imipramine improved learning and memory impairments, contrary to the findings of clinical meta-analyses showing that tricyclic antidepressants have no effect on cognitive dysfunction in depression [7,8]. In this study, we developed an ACMS model with imipramine-resistant properties to evaluate the effects of the DOR inverse agonist SYK-623. SYK-623 attenuated depression-like behavior and working and short-term memory impairments in an ACMS mouse model. 

Inverse agonists not only suppress constitutive receptor activity but also act as an antagonist against endogenous ligands [18]. In the present study, we found that NTI, a neutral antagonist, had no effect on ACMS-induced impairments but rather tended to worsen depressive-like behavior. In our previous study, SYK-623 but not NTI significantly improved spatial working memory impairment. Moreover, the effects of SYK-623 were inhibited by the pre-administration of NTI [17]. Thus, the effects of SYK-623 are probably based on the inhibition of ligand-independent constitutive receptor activity rather than antagonism of the endogenous ligand.

Astrocyte loss is implicated in the pathophysiology of depression, and the protection of astrocytes by antidepressants, such as imipramine and fluoxetine, is thought to contribute to antidepressant effects [22,33]. Although DOR has been implicated in astrocyte survival and proliferation [38,39], SYK-623 did not suppress imipramine-resistant astrocyte loss in ACMS mice. This finding suggests that astrocytes are not the targets of the antidepressant effects of DOR inverse agonists. 

Astrocytes play an important role in learning and memory [40]. Astrocyte dysfunction caused by the expression of tetanus neurotoxin impairs long-term memory but not working memory, suggesting that astrocytes play a more crucial role in long-term memory [41]. Similarly, neurogenesis is essential for long-term memory, and ablation leads to impaired long-term memory function [42]. In the present study, SYK-623 inhibited the ACMS-induced reduction in neurogenesis but not the loss of astrocytes or long-term memory impairment. These results suggest that, for long-term memory to be effectively established, treatments need to address both astrocytes and neurogenesis. 

Neurogenesis is thought to play an important role in antidepressant action, since antidepressants, including imipramine, increase neurogenesis, and the inhibition of neurogenesis attenuates the effects of antidepressants [43,44,45,46,47]. In contrast, in the present study, imipramine did not increase neurogenesis in ACMS mice, which is consistent with previous results using a repeated ACTH model [48]. Thus, although a reduction in neurogenesis was observed regardless of imipramine sensitivity, the suppression of imipramine-induced neurogenesis may underlie the development of imipramine resistance in the ACMS model. Chronic administration of ACTH suppresses imipramine-induced increases in 5-HT and the expression of brain-derived neurotrophic factor (BDNF), an important factor for the survival and maturation of newborn neurons, and increases the sensitivity of 5-HT_2A_ receptors, a negative regulator of neurogenesis. Abnormalities in monoaminergic signaling pathways may contribute to a reduction in the effects of imipramine on neurogenesis. Despite the treatment-resistant properties of the ACMS model, SYK-623 suppressed the decrease in neurogenesis. These findings suggest that SYK-623 protects against neurogenesis and prevents the induction of imipramine-resistant depression-like behavior in ACMS mice via mechanisms distinct from those of imipramine. 

GABAergic neurons have been shown to be involved in the process of neurogenesis, and in particular, the activation of PV-GABA neurons in the dentate gyrus is required for differentiation from neural progenitors to neurons and survival [48,49,50]; PV-GABA neurons are vulnerable to stress, and their abnormalities in the hippocampus induce the reduction of neurogenesis and behavioral impairments associated with psychiatric disorders [24,38,39]. Since DORs are abundantly expressed in PV-GABA neurons in the hippocampus [51], we predicted that PV-GABA neurons would be one of the main targets of SYK-623. SYK-623 inhibited ACMS-induced reductions in PV-GABA neuronal density and GAD67. Thus, SYK-623 may protect against neurogenesis by maintaining GABA neuronal survival under chronic stress conditions. 

PV-GABA neurons play an important role in hippocampal working memory. Selective ablation of NMDA receptors in hippocampal PV-GABA neurons induces working memory impairment [52]. Conversely, enhanced PV-GABAergic innervation improves working memory [53]. Although PV-GABA neurons regulate neurogenesis [54,55], the ablation of neurogenesis does not impair working memory [56,57], indicating that neurogenesis may not contribute to working memory. These findings suggest that the reduction in PV-GABA neurons in ACMS mice may underlie the impairment of working memory independent of the decrease in neurogenesis and that the protective effects of SYK-623 on cognitive function may be critically dependent on the preservation of PV-GABA neurons.

Although imipramine prevents working memory impairments in CMS animal models [58,59], its effects on PV-GABAergic impairment remain unknown. In addition, the present study did not examine the effect of imipramine on the decrease in PV-GABA neurons in the ACMS model. This has not been clarified for other antidepressants and should be investigated in the future with various antidepressants.

Cognitive impairments in major depressive disorder include deficits in working memory, attention, executive function, and processing speed, which may contribute to depressive mood and anhedonia [60]. In a meta-analysis of treatment effects, antidepressants, except vortioxetine, did not significantly improve cognitive function [7,61]. As residual cognitive impairment after mood remission can exacerbate depressive symptoms, effective drugs are required. Working memory is fundamental to the performance of many cognitive tasks and daily activities, and deficits in working memory are at the top of endophenotype candidates for recurrent major depressive disorder [62]. In the present study, SYK-623 protected working and short-term memory functions in ACMS mice, in addition to its previously reported effects on spatial working memory [16]. The pharmacological characteristics of SYK-623 suggest its potential use as a novel antidepressant with cognition-enhancing properties.

In contrast to the chronic SYK-623 administration results, chronic treatment of DOR agonists showed antidepressant and neuroprotective effects [63,64,65,66]. In these studies, the experimental conditions, such as the models and animals, are different from those in the present study. Chronic treatment of SNC80 or KNT-127 showed antidepressant-like effects in a hyperemotional response of olfactory bulbectomized Wistar rats [63,64]. KNT-127 showed antidepressant-like effects, as assessed by social interaction and neuroprotective effects in compensatory social defeat stress mice [65]. Moreover, chronic administration of SNC-121 protected the retinal ganglion neurons of glaucoma model mice, though not a model of depression [66]. Therefore, these differences in experimental conditions may be the cause of contradictions between SYK-623 and DOR agonists. Alternatively, we propose two hypotheses from a pharmacological perspective. The first hypothesis is that DOR agonists and inverse agonists exhibit antidepressant-like and neuroprotective effects via distinct intracellular signals. SYK-623 has been shown to elevate intracellular cAMP levels by inhibiting the constitutive activity of Gi-protein-coupled DORs [67]. Furthermore, cAMP and its downstream signaling pathways have been implicated in the mechanisms underlying antidepressant effects, including the promotion of neurogenesis and the protection of neurons from the effects of chronic stress [68,69,70]. In contrast, recent studies on the DOR agonist KNT-127 have reported that its antidepressant effects involve the enhancement of excitatory neurotransmission in the cortex via the phosphoinositide-3-kinase/Akt pathway [65,71]. This signaling pathway also mediated the neuroprotective effects of SNC-121 and biphalin in the retinal ganglion of glaucoma models and in the brain of a neonatal cerebral ischemia model, respectively [66,72]. Although the mechanism of signal activation by DOR has not been clearly identified [73], it is possible that DOR agonists exert their effects via a signal distinct from that of the cAMP pathway. The second hypothesis is that SYK-623 potentiates the action of endogenous agonists through its receptor-upregulating effects, a commonly observed property of inverse agonists [18]. For example, the histamine H_2_-blockers cimetidine and ranitidine, which act as inverse agonists, cause a treatment period-dependent decrease in gastric acid inhibition, which is produced by an increase in H_2_ receptors [18]. The detailed mechanism of action of SYK-623 remains to be investigated in future studies.

## 5. Conclusions

In the present study, we showed that the DOR inverse agonist SYK-623 prevented the induction of imipramine-resistant depression-like behavior and memory impairment in ACMS mice. These effects may involve the protection of neurogenesis, including that of PV-GABAergic neurons. These findings suggest that DOR inverse agonists may be novel antidepressants with cognition-improving properties. The pathogenesis of depression involves many different mechanisms, such as abnormal synaptic plasticity, and many regions, such as the cortex and nucleus accumbens [74]. Further investigation into the details of SYK-623 in these mechanisms may reveal more effective therapeutic targets.

## Figures and Tables

**Figure 1 jcm-13-00608-f001:**
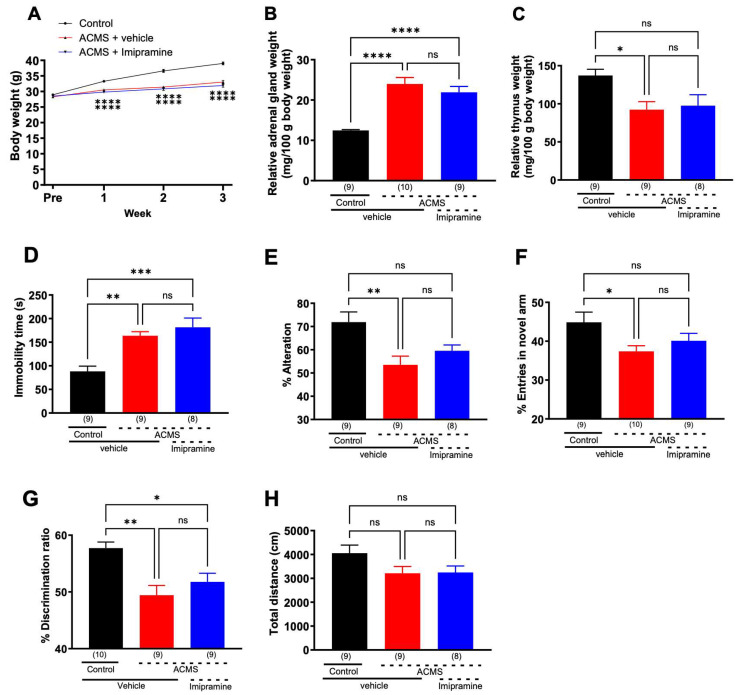
Effects of imipramine on ACMS-induced physical and behavioral changes in mice: (**A**) body weight, (**B**) relative adrenal gland weight, (**C**) relative thymus gland weight, (**D**) immobility time in the TST, (**E**) % alternations in the Y-maze test, (**F**) % entries in novel arm in the modified Y-maze test, (**G**) % discrimination ratio in the novel object recognition test, (**H**) total distance in the open-field test. Each point represents the mean ± SEM. ns: non-significant, * *p* < 0.05, ** *p* < 0.01, *** *p* < 0.001, **** *p* < 0.0001 ((**A**): Two-way repeated measures ANOVA followed by Holm-Sidak’s multiple comparison test; (**B**–**H**): One-way ANOVA followed by Holm-Sidak’s multiple comparison test). The number in parentheses represents the number of mice.

**Figure 2 jcm-13-00608-f002:**
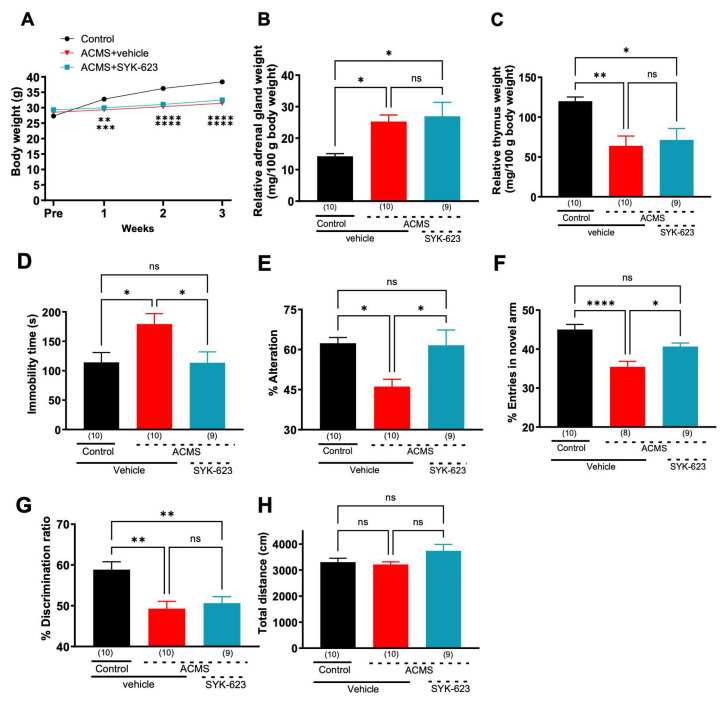
Effects of SYK-623 on ACMS-induced physical and behavioral changes in mice: (**A**) body weight, (**B**) relative adrenal gland weight, (**C**) relative thymus gland weight, (**D**) immobility time in the TST, (**E**) % alternations in the Y-maze, (**F**) % entries in novel arm in the modified Y-maze, (**G**) % discrimination ratio in the novel object recognition test, (**H**) total distance in the open-field test. Each point represents the mean ± SEM. ns: non-significant, * *p* < 0.05, ** *p* < 0.01, *** *p* < 0.001, **** *p* < 0.0001 ((**A**): Two-way repeated measures ANOVA; (**B**–**H**): One-way ANOVA followed by Holm-Sidak’s multiple comparison test). The number in parentheses represents the number of mice.

**Figure 3 jcm-13-00608-f003:**
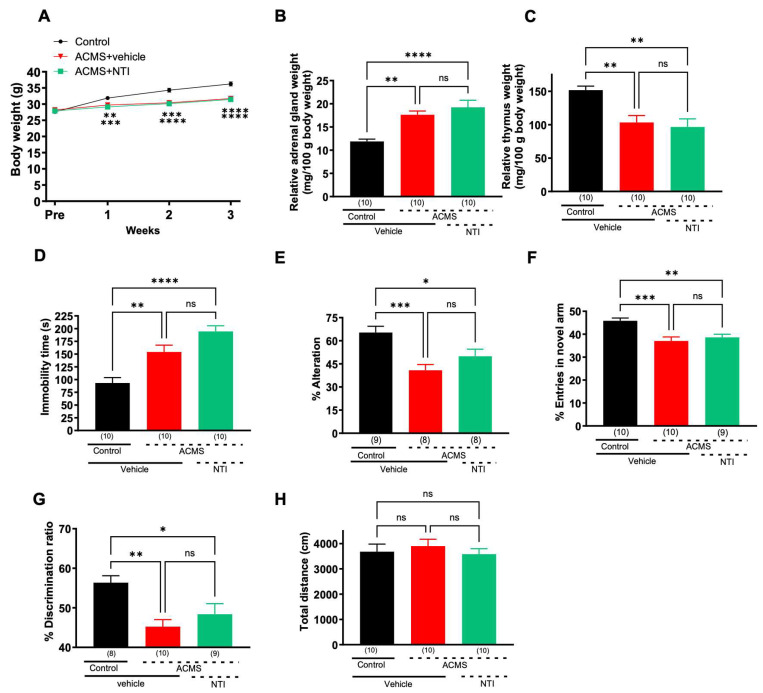
Effects of NTI on ACMS-induced physical and behavioral changes in mice: (**A**) body weight, (**B**) relative adrenal gland weight, (**C**) relative thymus gland weight, (**D**) immobility time in the tail suspension test, (**E**) % alternations in the Y-maze, (**F**) % entries in novel arm in the modified Y-maze, (**G**) % discrimination ratio in the novel object recognition test, (**H**) total distance in the open-field test. Each point represents the mean ± SEM. ns: non-significant, * *p* < 0.05, ** *p* < 0.01, *** *p* < 0.001, **** *p* < 0.0001 ((**A**): Two-way repeated measures ANOVA; (**B**–**H**): One-way ANOVA followed by Holm-Sidak’s multiple comparison test). The number in parentheses represents the number of mice.

**Figure 4 jcm-13-00608-f004:**
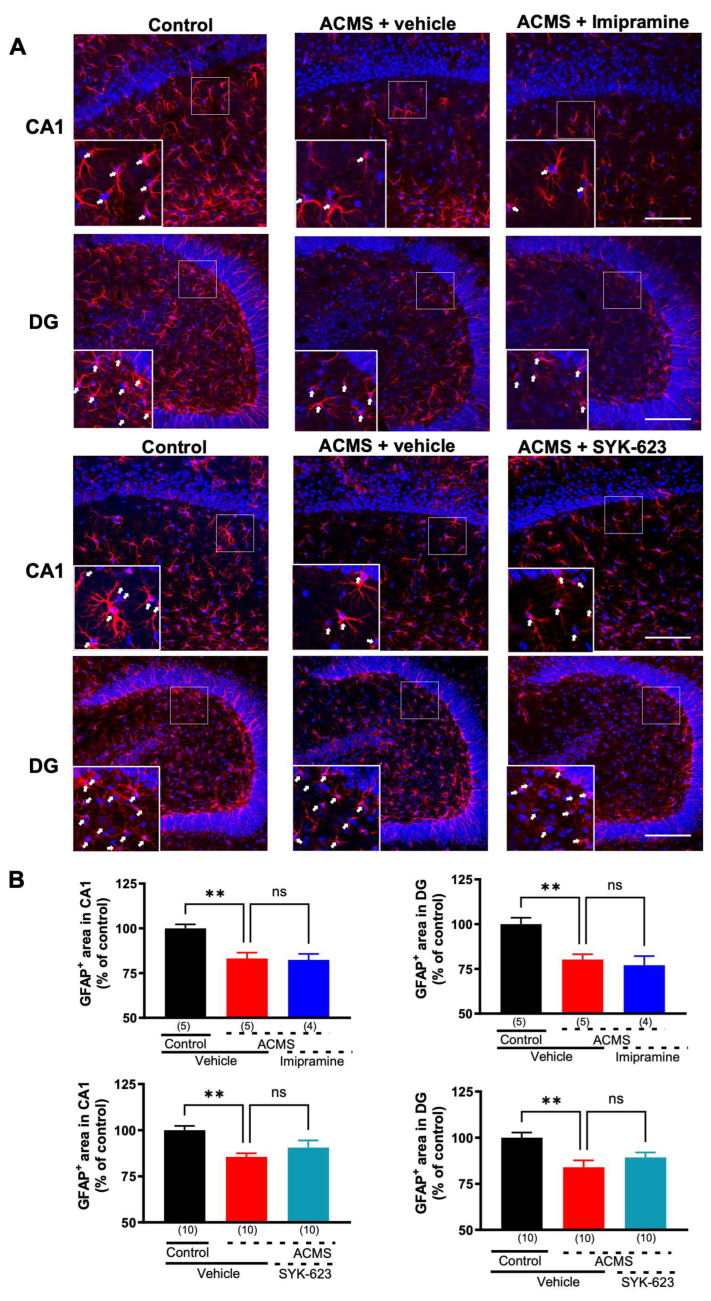
Effects of imipramine or SYK-623 on the ACMS-induced reduction of the GFAP-positive area. (**A**) Representative immunofluorescence images of GFAP immunoreactivity (red) and nuclei stained with DAPI (blue) in hippocampal CA1 and the dentate gyrus. The area enclosed by the white square was enlarged and superimposed on the lower left corner of each image. White arrows indicate GFAP-positive cells. (**B**) Summary of the GFAP-positive area in the hippocampal dentate gyrus. Each point represents the mean ± SEM. ns: non-significant, ** *p* < 0.01 (One-way ANOVA followed by Holm-Sidak’s multiple com-parison test). The number in parentheses represents the number of mice. Scale bar = 100 μm.

**Figure 5 jcm-13-00608-f005:**
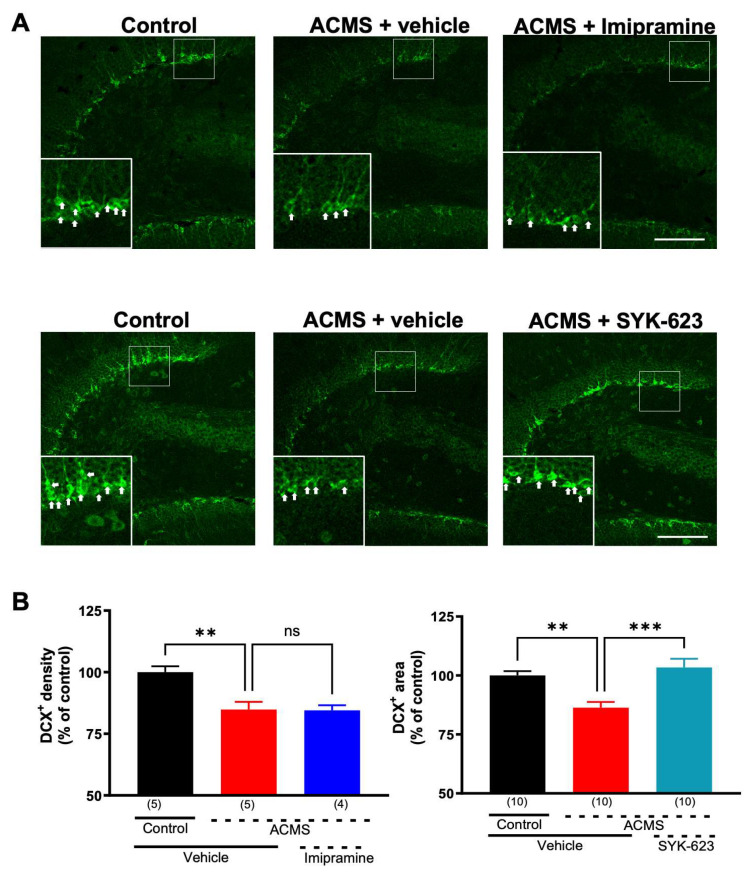
Effects of imipramine and SYK-623 on the ACMS-induced reduction of the DCX-positive area. (**A**) Representative immunofluorescence images of DCX immunoreactivity (green) in the hippocampal dentate gyrus. The area enclosed by the white square was enlarged and superimposed on the lower-left corner of each image. White arrows indicate DCX-positive cells. (**B**) Summary of DCX-positive areas in the hippocampal dentate gyrus. Each point represents the mean ± SEM. ** *p* < 0.01, *** *p* < 0.001 (One-way ANOVA followed by Holm-Sidak’s multiple comparison test). The numbers in parentheses represent the number of mice. Scale bar = 100 μm.

**Figure 6 jcm-13-00608-f006:**
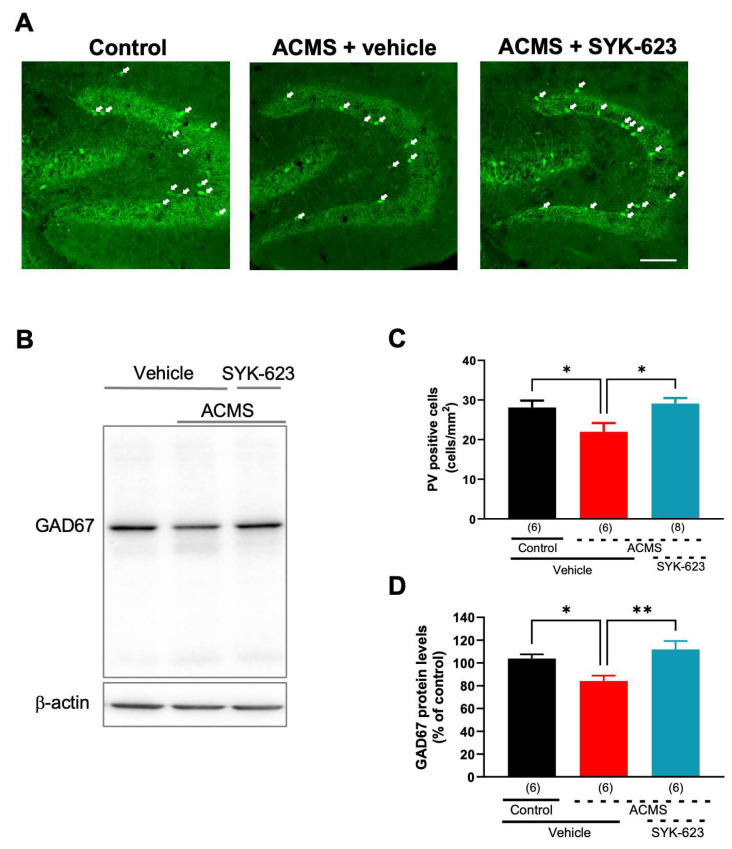
Effects of SYK-623 on ACMS-induced GABAergic impairment. (**A**) Representative immunofluorescence images of PV immunoreactivity (green) in the hippocampal dentate gyrus. White arrows indicate PV-positive cells. (**B**) Representative western blotting of the GAD67 protein levels within hippocampal tissue. (**C**) Summary of PV-positive cell density in the hippocampal dentate gyrus. (**D**) Semi-quantitative analysis of GAD67 protein levels in the hippocampus. GAD67 protein levels were normalized to β-actin protein expression. Each point represents the mean ± SEM. * *p* < 0.05, ** *p* < 0.01 (One-way ANOVA followed by Holm-Sidak’s multiple comparison test). The number in parentheses represents the number of mice. Scale bar = 100 μm.

**Table 1 jcm-13-00608-t001:** Experimental schedule.

Days	Drug Treatment	Short Stressor (9:30–14:00)	Overnight Stressor
1	ACTH + Vehicle, Imipramine, SYK-623, or NTI	Restraint stress	Night lighting
2		Forced swimming	Cage tilting
3		Restraint stress	Bed deprivation
4			Wet bedding
5		Forced swimming	Water deprivation
6			
7		Restraint stress	Cage tilting
8		Forced swimming	
9		Restraint stress	Bed deprivation
10			Wet bedding
11		Restraint stress	Food deprivation
12		Forced swimming	Cage tilting
13			
14		Forced swimming	Cage tilting
15		Restraint stress	Wet bedding
16		Forced swimming	Night lighting
17		Restraint stress	Cage tilting
18		Forced swimming	Bed deprivation
19		Restraint stress	Night lighting
20			
21			Cage tilting
22			Lighting
23		Modified Y-maze test *	Bed deprivation
24		Open-field test *	Wet bedding
25		Y-maze test *	Night lighting
26		Restraint stress	Cage tilting
27		Novel object recognition test *	
28		Novel object recognition test *	Night lighting
29		Tail suspension test *	Bed deprivation
30		Tail suspension test *	Cage tilting
31			Wet bedding

* Behavioral experiments were conducted between 10:00 and 19:00.

## Data Availability

Data have been presented in the manuscript. Additional raw data are available from the corresponding author upon request.

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
