# Peer review of "SYK-623, a δ Opioid Receptor Inverse Agonist, Mitigates Chronic Stress-Induced Behavioral Abnormalities and Disrupted Neurogenesis"

_jcm, 2024, doi:10.3390/jcm13020608_

Round 1

Reviewer 1 Report

Comments and Suggestions for Authors
    • The introduction provides a clear overview of the study, highlighting the significance of the delta-opioid receptor (DOR) and the synthesized compounds (SYK-657 and SYK-623). However, a few things need to be addressed in the manuscript.
    • 1- It might be helpful to include a brief definition or explanation of terms such as "inverse agonists" for readers who may not be familiar with pharmacological terminology.

    • 2- Briefly discuss why this particular strain was chosen over others and if any known characteristics might impact the study.

    3- Provide more details on the controlled environment in which the mice were kept. Specify if there were any variations in environmental conditions during the experiment that could potentially impact the results.
    4- Justify the timing and frequency of the stressors and the ACTH administration.
    5- Discuss the rationale behind choosing to replace one specific object in the novel object recognition test. What criteria were used to select the novel object?
    6- Highlight the significance of SYK-623's ability to attenuate ACMS-induced behavioral changes. Discuss potential mechanisms behind this protective effect.
Comments on the Quality of English Language

Minor editing of English language required

Author Response

Dear Reviewer 1,

 We express our gratitude for the positive evaluation of our research. We have listed a response to each comment in a point-by-point manner below.

 1- It might be helpful to include a brief definition or explanation of terms such as "inverse agonists" for readers who may not be familiar with pharmacological terminology.

Reply: We revised the explanation regarding inverse agonists as follows (Introduction, line 57): “Receptors may have intrinsic activity even in the absence of agonist binding. Agonists bind to receptors and increase their activity, whereas inverse agonists decrease the intrinsic activity of receptors (negative intrinsic activity) [18]]”

2- Briefly discuss why this particular strain was chosen over others and if any known characteristics might impact the study.

Reply: We described the reason for choosing ddY-strain mice as follows (2.1. Subjects, line 85): “Six-week-old male ddY mice (Japan SLC, Shizuoka, Japan) were used for all the experiments, since we previously reported that repeated administration of ACTH causes imipramine resistance in ddY mice [21]. “

3- Provide more details on the controlled environment in which the mice were kept. Specify if there were any variations in environmental conditions during the experiment that could potentially impact the results.

Reply: We added the following information regarding the experimental environment (2.1. Subjects, line 91): “Mice were housed at 4-5 mice per cage (30×20×13 cm). Bedding material was made of paper (paperclean®, Japan SLC) and was replaced after the wet bedding stress of ACMS (Table 1). To avoid the impact of various environmental changes on the mice, experiments were conducted in a dedicated soundproof room until the completion of the behavioral experiments.”

4- Justify the timing and frequency of the stressors and the ACTH administration.

Reply: We revised the manuscript as follows (2.2. ACMS Model Mice, line 98): “Individual stressors in CMS are exposed to animals at various timings and methods, depending on the laboratory [25]. To avoid excessive impairments induced by the combination of ACTH, we used a CMS protocol, which did cause moderate weight loss in the preliminary experiments (data not shown). To elevate low corticosterone levels in the light phase, ACTH (Cortrosyn, 0.45 mg/kg, Daiichi Sankyo, Tokyo, Japan) was subcu-taneously administered once daily at 9:30-11:30 based on previous reports [21].”

5- Discuss the rationale behind choosing to replace one specific object in the novel object recognition test. What criteria were used to select the novel object?

 Reply: We added the following explanation about the two objects and the criteria for their selection (2.7. Novel Object Recognition Tests, line 157): “The two objects are plastic toys in which the mice showed equal interest in the preliminary study.”

6- Highlight the significance of SYK-623's ability to attenuate ACMS-induced behavioral changes. Discuss potential mechanisms behind this protective effect.

Reply: We revised the discussion as follows (Discussion, line 436): ” Working memory is fundamental to the performance of many cognitive tasks and daily activities, and deficits in working memory are at the top of endophenotype candidates for recurrent major depressive disorder [62]. In the present study, SYK-623 protected working and short-term memory functions in ACMS mice, in addition to its previously reported effects on spatial working memory [16]. The pharmacological characteristics of SYK-623 suggest its potential use as a novel antidepressant with cognition-enhancing properties.”

As commented by the reviewer 1, the description of the neuroprotective mechanism was insufficient. In the reviewed manuscript, the mechanism of antidepressant-like efects of SYK-623 was described in discussion. Therefore, to avoid duplication with this description, we revised it to include the mechanism of neuroprotection and moved this paragraph to the latter part of the discussion. The revised text is as follows (Discussion, line 452): " Alternatively, we propose two hypotheses from a pharmacological perspective. The first hypothesis was that DOR agonists and inverse agonists exhibit antidepressant-like and neuroprotective effects via distinct intracellular signals. SYK-623 has been shown to elevate intracellular cAMP levels by inhibiting the constitutive activity of Gi-protein-coupled DORs [67]. Furthermore, cAMP and its downstream signaling pathways have been implicated in the mechanisms underlying antidepressant effects, including the promotion of neurogenesis and the protection of neurons from the effects of chronic stress [68–70]. In contrast, recent studies on the DOR agonist KNT-127 have re-ported that its antidepressant effects involve the enhancement of excitatory neurotransmission in the cortex via the phosphoinositide-3-kinase/Akt pathway [65,71]. This signaling pathway also mediated the neuroprotective effects of SNC-121 and biphalin in the retinal ganglion of glaucoma models and in the brain of neonatal cerebral ischemia model, respectively [66,72]. Although the mechanism of signal activation by DOR has not been clearly identified [73], it is possible that DOR agonists exert their effects via a signal distinct from that of the cAMP pathway. “

Reviewer 2 Report

Comments and Suggestions for Authors

In this manuscript, the authors investigated the effects of the DOR inverse agonist SYK-623 on mild chronic stress (ACMS)-induced physical, behavioral, and changes in astrocyte expression and neurogenesis markers. Their results showed that SYK-623 prevented the induction of imipramine-resistant depression-like behavior and memory impairment in ACMS mice. This protective effect may be due to protection of neurogenesis in the hippocampus. Results from this study suggest the importance of DOR receptor modulators as a potential therapeutic target for chronic mild stress induced depression. Overall, the methods were sound, results are novel, and discussion was well presented. However, here are my comments/concerns that need to be addressed, I divided them by section:

The abstract:

1-     ACHT was introduced for the first time in the abstract (line 18) and it should have been abbreviated at its first appearance in the abstract as well as in the body of the manuscript.

2-     ACMS is the abbreviation of chronic mild stress exposure and was first introduced at line 18. The “A” in ACMS stands for what? Why is it not CMS instead? It was also mentioned twice in the text while referring to published work in the literature in the manuscript (line 53, 268). Please clarify if there are differences between the two models or use a consistent term throughout the manuscript instead.

3-     In the last sentence of the abstract says “ …. disturbs neurogenesis caused by chronic stress”. I think you meant disturbed, reduced, or decreased neurogenesis. Please correct that since it completely changed the conclusion.

Introduction:

1-      I’d recommend start talking about chronic stress and depression then talking about DOR receptor and its ligands. (Shift paragraph 3 to be the 1st, then make paragraphs 1 and 2 the 2nd and 3rd paragraphs).

2-      I’d recommend adding 1 sentence at the end of the 2nd paragraph (line 37) to explain why DOR ligands have not been developed clinically. Is it due to their safety profile? Have any clinical trials using them started and stopped in the past?

3-      I prefer adding a few sentences about neurogenesis and GABAergic neurons and how they are affected by chronic stress in the introduction section before introducing them later in the manuscript.

4-      I’d recommend not referring to result figures in the introduction section (Fig 1 line 68). Instead, I would recommend replacing that by saying as will be discussed later in the manuscript.

Materials and Methods:

1-      Table 1, column 2: Did you mean ACHT + one of these options? If this is the case, please add commas between options and "or" before NIT.

2-      Line 99: it says diluted 10 times. Diluted by what? Water or saline?

3-      Following on the previous comment, does that mean that different vehicles were used for your different studies. This can justify using different ACMS groups treated with vehicle in SYK-623 and NTI studies. If not, I’d recommend adding a sentence explaining the reason behind using different control and vehicle groups for the different drugs studies. Otherwise, it would be easy to add the NTI treated group and comparing all groups together and saving a lot of animals.

4-      I’d recommend adding a table including all groups used in the studies and what treatments they received (SYK-623, NTI,…) and the number of mice in each group to help understanding the results later. Also, you might add how many mice brains were fixed or frozen from each group for the different experiments you conducted.

5-      It seems that symbols for (micron, beta, … etc) were all replaced by the same different symbol (lines 148, 159, 172, 326). Please use the correct symbols.

6-      In the statistics section. Please include all statistical tests performed to analyze the data, I feel it's not enough to mention them in the legends of the figures. Please also explain why you used Sidak’s posthoc test. Was that the test recommended by the software?

Discussion:

1-      Please modify the first sentence in line 340 to make it more clear. It sounds like it says that antagonists are a type of inverse agonists.

2-      On the paragraph starts on line 348, can you please mention what tests for depression and anxiety-like behaviors used for the studies you cited. Anxiety-like behaviors evaluation in rodents are assay dependent and the results might differ depending on the time point after stressors exposure at which the test was performed. You may discuss that a little bit further because this might explain why your present results contradict those previous reports.

Author Response

Dear Reviewer 2,

 We express our gratitude for the positive evaluation of our research. We have listed a response to each comment in a point-by-point manner below. Supplemental  table is provided only in the attachment file.  

The abstract:

1. ACHT was introduced for the first time in the abstract (line 18) and it should have been abbreviated at its first appearance in the abstract as well as in the body of the manuscript.

Reply: We revised the sentence of abstract as follows (Abstract, line 18): “Male ddY mice received daily treatment of adrenocorticotropic hormone (ACTH) combined with chronic mild stress exposure (ACMS).”

2.  ACMS is the abbreviation of chronic mild stress exposure and was first introduced at line 18. The “A” in ACMS stands for what? Why is it not CMS instead? It was also mentioned twice in the text while referring to published work in the literature in the manuscript (line 53, 268). Please clarify if there are differences between the two models or use a consistent term throughout the manuscript instead.

Reply: Conventional CMS model is imipramine-sensitive and does not include ACTH treatment in its preparation protocols. We administered ACTH combined with CMS exposure to develop resistance to imipramine treatment; we abbreviated the combined treatment of ACTH and CMS as ACMS to distinguish it from the conventional CMS. To avoid misunderstandings, we have defined the ACMS abbreviation by adding the following sentence (Introduction, line 72): “We abbreviated the combined treatment of ACTH and CMS as ACMS to distinguish it from the conventional CMS.”

3. In the last sentence of the abstract says “ …. disturbs neurogenesis caused by chronic stress”. I think you meant disturbed, reduced, or decreased neurogenesis. Please correct that since it completely changed the conclusion.

Reply: We appreciate the reviewer for finding the error, and corrected the sentence as follow (Abstract, line 28): “Thus, SYK-623 mitigates behavioral abnormalities and disturbed neurogenesis caused by chronic stress.”

Introduction:

1. I’d recommend start talking about chronic stress and depression then talking about DOR receptor and its ligands. (Shift paragraph 3 to be the 1st, then make paragraphs 1 and 2 the 2ndand 3rd paragraphs).

Reply: In accordance with the reviewer's comments, the third paragraph was moved to the first paragraph.

2.  I’d recommend adding 1 sentence at the end of the 2nd paragraph (line 37) to explain why DOR ligands have not been developed clinically. Is it due to their safety profile? Have any clinical trials using them started and stopped in the past?

      Reply: We added a sentence as follows (introduction, line 51): “Several DOR agonists were evaluated in clinical trials for postherpetic neuralgia (NCT01058642)), post-molar extraction pain (NCT00993863), osteoarthritis of the knee (NCT00979953) and anxious depression (NCT00759395), but they did not demonstrate sufficient efficacy.”

3.  I prefer adding a few sentences about neurogenesis and GABAergic neurons and how they are affected by chronic stress in the introduction section before introducing them later in the manuscript.

      Reply: We added a few sentences as follows (Introduction, line 74) : “Chronic stress reduces astrocytes, neurogenesis, and parvalbumin (PV)-positive GABA neurons, which are crucial for the expression of learning, memory, and antidepressant effects [22–24]. Thus, to investigate the mechanism of the effects of SYK-623, we examined whether SYK-623 protects these cells against ACMS in the hippocampus.”

4.  I’d recommend not referring to result figures in the introduction section (Fig 1 line 68). Instead, I would recommend replacing that by saying as will be discussed later in the manuscript.

Reply: The sentence was revised as follows (Introduction, line 69): “Therefore, we demonstrated that the combination of repeated ACTH administration with CMS to induce depression-like behavior and cognitive decline resulted in symptoms of imipramine resistance as will be discussed later in the manuscript.”

Materials and Methods:

1. Table 1, column 2: Did you mean ACHT + one of these options? If this is the case, please add commas between options and "or" before NIT.

Reply: We have revised the table 1 according to the reviewers' comment.

2. Line 99: it says diluted 10 times. Diluted by what? Water or saline?

      Reply: SYK-623 stock solution was diluted by saline. We corrected the sentence as follow (2.3. Drugs, line 116): “At the time of use, it was diluted 10-fold by saline (Otsuka Pharmaceutical Co. Ltd., Tokyo, Japan).”

3. Following on the previous comment, does that mean that different vehicles were used for your different studies. This can justify using different ACMS groups treated with vehicle in SYK-623 and NTI studies. If not, I’d recommend adding a sentence explaining the reason behind using different control and vehicle groups for the different drugs studies. Otherwise, it would be easy to add the NTI treated group and comparing all groups together and saving a lot of animals.

Reply: The corresponding vehicle was used as a vehicle control for each drug. We added a sentence as follows (2.3. Drugs, line 121): “DMSO-containing (10%) saline and normal saline were administered as vehicle controls for SYK-623 and other drugs (imipramine and NTI), respectively.”

4.  I’d recommend adding a table including all groups used in the studies and what treatments they received (SYK-623, NTI,…) and the number of mice in each group to help understanding the results later. Also, you might add how many mice brains were fixed or frozen from each group for the different experiments you conducted.

Reply: In accordance with the reviewer's comments, we added a supplemental table 1, and a sentence as follows: “Treatment groupings were shown in Supplemental Table 1.”

5. It seems that symbols for (micron, beta, … etc) were all replaced by the same different symbol (lines 148, 159, 172, 326). Please use the correct symbols.

Reply: We appreciate the finding of our errors. The indicated symbols were corrected accordingly (2,9. Immunohistochemistry, line, 171, line 182; 2.10. SDS-PAGE, line 191, line 195; Figure 6, line 355).

6. In the statistics section. Please include all statistical tests performed to analyze the data, I feel it's not enough to mention them in the legends of the figures. Please also explain why you used Sidak’s posthoc test. Was that the test recommended by the software?

Reply: We apologize for the incorrect description of the statistical methods in our manuscript. The text has been corrected to 'Holm-Sidak’s multiple comparison test' in the legends. we have added all the statistical methods used and provided an explanation for choosing the Holm-Sidak’s method as follows (2.11. Statistics, line 205): " Two-way analysis of variance was performed for time course of body weight changes, and one-way analysis of variance was performed for behavioral experiments, immuno-histochemistry, and Western blotting. Holm-Sidak's method, which exhibits adequate type I error rate control and high power, was used for all multiple comparisons [31].”

Discussion:

1. Please modify the first sentence in line 340 to make it more clear. It sounds like it says that antagonists are a type of inverse agonists.

Reply: The sentence has been revised as follows (Discussion, line 369): “Inverse agonists not only suppress constitutive receptor activity but also act as an antagonist against endogenous ligands [18].”

2. On the paragraph starts on line 348, can you please mention what tests for depression and anxiety-like behaviors used for the studies you cited. Anxiety-like behaviors evaluation in rodents are assay dependent and the results might differ depending on the time point after stressors exposure at which the test was performed. You may discuss that a little bit further because this might explain why your present results contradict those previous reports.

Reply: Following the reviewer 2's comments, we have added descriptions regarding the experimental conditions of previous studies of DOR agonist. Since the present manuscript did not show the results regarding anxiolytic effects, we discussed the antidepressant effects. Additionally, reviewer 1 requested for a discussion of the neuroprotective effect, so a description of this topic has been added within this paragraph. We have revised the first half of the paragraph as follows (Discussion, line 443):” In contrast to the chronic SYK-623 administration results, chronic treatment of DOR agonists showed antidepressant and neuroprotective effects [63–66]. In these studies, experimental conditions, such as the models and animals, are different from those in the present study. Chronic treatment of SNC80 or KNT-127 showed antidepressant-like effects in a hyperemotional response of olfactory bulbectomized Wistar rats [63,64]. KNT-127 showed antidepressant-like effects as assessed by social interaction and neuroprotective effects in compensatory social defeat stress mice [65]. Moreover, chronic administration of SNC-121 protected the retinal ganglion neurons of glaucoma model mice, though not a model of depression [66]. Therefore, these differences in experimental conditions may be the cause of contradictions between SYK-623 and DOR agonists. Alternatively, we propose two hypotheses from a pharmacological perspective. The first hypothesis was that DOR agonists and inverse agonists exhibit antidepressant-like and neuroprotective effects via distinct intracellular signals.”
